# Broadly Effective ACE2 Decoy Proteins Protect Mice from Lethal SARS-CoV-2 Infection

Mengjia Lu,[a,b] Weitong Yao,[d] Yujun Li,[e] Danting Ma,[f] Zhaoyong Zhang,[c] Haimin Wang,[b*] Xiaojuan Tang,[a,b] Yanqun Wang,[c] Chao Li,[b] Dechun Cheng,[b] Hua Lin,[g] Yandong Yin,[a,b] Jincun Zhao,[c] Guocai Zhong[a,b,§,◇,∞]

aState Key Laboratory of Chemical Oncogenomics, Guangdong Provincial Key Laboratory of Chemical Genomics, Peking University Shenzhen Graduate School, Shenzhen, Guangdong, China

bShenzhen Bay Laboratory, Shenzhen, Guangdong, China

cState Key Laboratory of Respiratory Disease, Guangzhou Institute of Respiratory Health, First Affiliated Hospital of Guangzhou Medical University, Guangzhou, Guangdong, China

dHubei JiangXia Laboratory, Wuhan, Hubei, China

eShenzhen University School of Medicine, Shenzhen, Guangdong, China

fTianjin Medical University Chu Hsien-I Memorial Hospital, Tianjin, China

gBiomedical Research Center of South China, Fujian Normal University, Fuzhou, Fujian, China

Mengjia Lu, Weitong Yao, Yujun Li, Danting Ma, and Zhaoyong Zhang contributed equally to this work. Author order was determined by drawing straws.

**ABSTRACT** As severe acute respiratory syndrome coronavirus 2 (SARS-CoV-2) variants have been causing increasingly serious drug resistance problem, development of broadly effective and hard-to-escape anti-SARS-CoV-2 agents is an urgent need. Here, we describe further development and characterization of two SARS-CoV-2 receptor decoy proteins, ACE2-Ig-95 and ACE2-Ig-105/106. We found that both proteins had potent and robust *in vitro* neutralization activities against diverse SARS-CoV-2 variants, including BQ.1 and XBB.1, that are resistant to most clinically used monoclonal antibodies. In a stringent lethal SARS-CoV-2 infection mouse model, both proteins lowered the lung viral load by up to ~1,000-fold, prevented the emergence of clinical signs in >75% animals, and increased the animal survival rate from 0% (untreated) to >87.5% (treated). These results demonstrate that both proteins are good drug candidates for protecting animals from severe COVID-19. In a head-to-head comparison of these two proteins with five previously described ACE2-Ig constructs, we found that two constructs, each carrying five surface mutations in the ACE2 region, had partial loss of neutralization potency against three SARS-CoV-2 variants. These data suggest that extensively mutating ACE2 residues near the receptor binding domain (RBD)-binding interface should be avoided or performed with extra caution. Furthermore, we found that both ACE2-Ig-95 and ACE2-Ig-105/106 could be produced to the level of grams per liter, demonstrating the developability of them as biologic drug candidates. Stress condition stability testing of them further suggests that more studies are required in the future to improve the stability of these proteins. These studies provide useful insight into critical factors for engineering and preclinical development of ACE2 decoys as broadly effective therapeutics against diverse ACE2-utilizing coronaviruses.

**IMPORTANCE** Engineering soluble ACE2 proteins that function as a receptor decoy to block SARS-CoV-2 infection is a very attractive approach to creating broadly effective and hard-to-escape anti-SARS-CoV-2 agents. This article describes development of two antibody-like soluble ACE2 proteins that broadly block diverse SARS-CoV-2 variants, including Omicron. In a stringent COVID-19 mouse model, both proteins successfully protected >87.5% animals from lethal SARS-CoV-2 infection. In addition, a head-to-head comparison of the two constructs developed in this study with five previously described ACE2 decoy constructs was performed here. Two previously described constructs with relatively more ACE2 surface mutations were found with

Address correspondence to Guocai Zhong, guocai.zhong@umassmed.edu, Jincun Zhao, zhaojincun@gird.cn, Yandong Yin, yinyd@szbl.ac.cn, or Yujun Li, liyujun@szu.edu.cn.

*Present address: Haimin Wang, Horae Gene Therapy Center, University of Massachusetts Chan Medical School, Worcester, Massachusetts, USA.

§Present address: Guocai Zhong, Horae Gene Therapy Center, University of Massachusetts Chan Medical School, Worcester, Massachusetts, USA.

◇Present address: Guocai Zhong, RNA Therapeutics Institute, University of Massachusetts Chan Medical School, Worcester, Massachusetts, USA.

∞Present address: Guocai Zhong, Department of Biochemistry and Molecular Biotechnology, University of Massachusetts Chan Medical School, Worcester, Massachusetts, USA.

The authors declare no conflict of interest.

less robust neutralization activities against diverse SARS-CoV-2 variants. Furthermore, the developability of the two proteins as biologic drug candidates was also assessed here. This study provides two broad anti-SARS-CoV-2 drug candidates and useful insight into critical factors for engineering and preclinical development of ACE2 decoys as broadly effective therapeutics against diverse ACE2-utilizing coronaviruses.

**KEYWORDS** SARS-CoV-2, variant of concern, Omicron, ACE2, receptor decoy, ACE2-Ig, mouse model, lethal infection

The coronavirus disease 2019 (COVID-19) pandemic, which is caused by the severe acute respiratory syndrome coronavirus 2 (SARS-CoV-2), has triggered unprecedentedly rapid development of a number of countermeasures against COVID-19, including multiple prophylactic vaccines and a number of convalescent patient-derived monoclonal antibodies in clinical use (1, 2). In spite of these great achievements, SARS-CoV-2 has caused more than 600 million confirmed infections and over 6.5 million documented deaths, and the pandemic is still ongoing. This is because of the continuous emergence of new SARS-CoV-2 variants. Since late 2020, five rapidly spreading and immune-evasive variants of concern (VOCs [Alpha, Beta, Gamma, Delta, and Omicron]) have sequentially caused multiple waves of global transmission and infections, because each of these major VOCs has more and more amino acid substitutions that have affected transmissibility and sensitivity to infection- or vaccine-induced neutralizing antibodies (3).

SARS-CoV-2 utilizes ACE2 as a key cellular receptor to infect cells (4). The receptor binding domain (RBD) of the viral Spike protein is responsible for the interaction and binds ACE2 with high affinity (5). Antibodies targeting the interactions between ACE2 and SARS-CoV-2 Spike RBD efficiently neutralize SARS-CoV-2 infection and reduce viral load in animal models and COVID-19 patients (6–10). So far, there are 12 anti-SARS-CoV-2 monoclonal antibodies and four two-antibody cocktails approved for clinical use (11, 12). These antibody therapeutics offer a treatment option for individuals with severe COVID-19 and are especially important for high-risk individuals where vaccination is not very effective. However, the continuous emergence of SARS-CoV-2 variants with more and more mutations in the Spike RBD region has been causing increasingly serious drug resistance issues. The original Omicron variant BA.1, first detected in November 2021, has been found to have great resistance to the majority of the approved anti-SARS-CoV-2 antibody therapeutics, including 10 monoclonal antibodies and three antibody cocktails (12). Since then, monoclonal antibodies capable of neutralizing the original Omicron variant have been found to be largely inactive against the latest new Omicron BQ and XBB subvariants, which are currently causing most new infections (12, 13). This makes the antibody therapeutics once very useful for high-risk individuals (e.g., the elderly and immunocompromised) not a good option for these individuals now. In addition, SARS-CoV-2 has been found easy to develop resistance to small molecule inhibitors such as remdesivir and nirmatrelvir (14, 15). Therefore, the development of broadly effective and hard-to-escape anti-SARS-CoV-2 agents is an urgent need.

The receptor decoy is a very promising strategy toward broadly antiviral therapeutics and has been previously applied to the development of very potent, exceptionally broad, and difficult-to-escape HIV-1 entry inhibitors (16–18). ACE2-Ig, a recombinant Fc fusion protein of soluble human ACE2 (hACE2), could function as a decoy to compete with cell surface ACE2 receptor and thus should broadly block entry of diverse SARS-CoV-2 variants and be difficult to escape. We previously described the development of improved ACE2-Ig proteins that potently neutralized the prototype SARS-CoV-2 *in vitro* (19, 20). We also demonstrated in an adenovirus (Ad)-hACE2-sensitized mouse model that one of our early ACE2-Ig constructs is both prophylactically and therapeutically active against SARS-CoV-2 infection *in vivo* (21). Here, we further optimized our ACE2-Ig constructs and assessed the *in vivo* efficacies of two optimized proteins against lethal SARS-CoV-2 infection in a stringent B6.Cg-Tg(K18-ACE2)2Prlmn/J (here, K18-hACE2) mouse model (22).

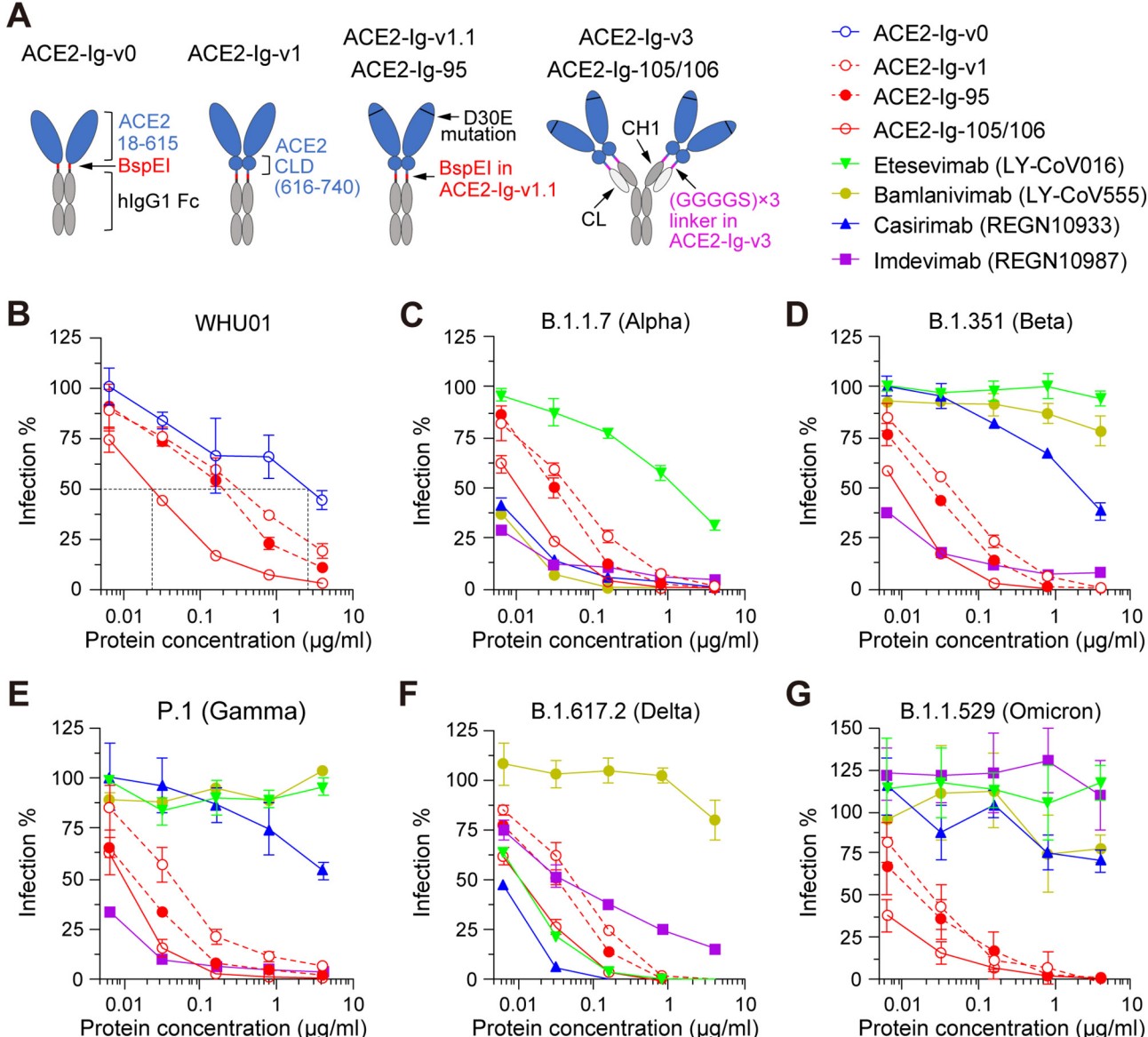

**FIG 1** ACE2-Ig-95 and ACE2-Ig-105/106 proteins but not monoclonal antibodies robustly neutralized pseudoviruses of diverse SARS-CoV-2 variants of concern. (A) Diagrams showing recombinant ACE2-Ig constructs characterized in the following studies. The non-self BspEI restriction site encoding three amino acids (Gly-Pro-Glu) was removed in ACE2-Ig-95, which also has a C-to-S mutation at position 5 of the hinge region. The non-self (GGGGS)×3 linker was completely removed in ACE2-Ig-105/106. CLD, collectrin-like domain; CH1, human IgG1 antibody heavy-chain constant domain 1; CL, human antibody κ light-chain constant domain. (B to G) The indicated ACE2-Ig constructs were compared with four previously approved anti-SARS-CoV-2 monoclonal antibodies for their *in vitro* neutralization potencies against pseudoviruses of six SARS-CoV-2 variants in HeLa-hACE2 cells, a stable cell line that overexpresses human ACE2. Pseudovirus infection-mediated luciferase reporter expression was measured at 48 h postinfection. Luciferase signals observed at each inhibitor concentration were divided by the signals observed at concentration zero to calculate percentage of infection values. The data shown are representative of three independent experiments performed by two different people with similar results, and data points represent mean ± SD from three biological replicates.

## RESULTS

**ACE2-Ig-95 and ACE2-Ig-105/106 showed robust and potent *in vitro* neutralization potency against pseudoviruses of diverse SARS-CoV-2 variants.** We previously described multiple ACE2-Ig constructs, ACE2-Ig-v0, ACE2-Ig-v1, ACE2-Ig-v1.1, and ACE2-Ig-v3 (Fig. 1A). ACE2-Ig-v0 is a homodimeric human ACE2 peptidase domain (amino acids [aa] 18 to 615) Fc fusion protein. ACE2-Ig-v1 carries both the peptidase domain and the Collectrin-like domain (CLD [aa 616 to 740]) of ACE2. ACE2-Ig-v1.1 is an ACE2 D30E mutant of ACE2-Ig-v1. ACE2-Ig-v3 is an antibody-like fusion protein wherein the Fv portions of both the heavy and light chains of human IgG1 have been replaced with the ACE2 portion

of the ACE2-Ig-v1.1 construct. Here, we slightly modified ACE2-Ig-v1.1 by removing three amino acids (Gly-Pro-Glu) encoded by a non-self BspEI restriction site between the ACE2 domain and the IgG1 hinge and then introducing a C-to-S mutation at position 5 of the hinge region to give the hinge more flexibility. The resulted new construct was named as ACE2-Ig-95 (Fig. 1A). Then, because residues 725 to 740 of the ACE2 ectodomain are unstructured and might be able to serve as a linker (23), we modified ACE2-Ig-v3 by gradually shortening the non-self (GGGGS)×3 linker between ACE2 and the $\kappa$ light-chain constant domain (CL), as well as between ACE2 and the first constant domain of the IgG1 heavy chain (CH1). Compared to ACE2-Ig-v3, a new construct that has the non-self (GGGGS)×3 linker completely removed showed similar neutralization potency but improved *in vivo* pharmacokinetics profiles, including increased initial plasma concentration and extended plasma half-life (see Fig. S1 in the supplemental material). This new antibody-like construct was named ACE2-Ig-105/106 and kept for further analysis (Fig. 1A). In an *in vitro* pseudovirus neutralization assay using a prototype SARS-CoV-2 (WHU01) pseudovirus, similar to what we observed in the previous study (20), ACE2-Ig-105/106 showed about 100-fold improvement over ACE2-Ig-v0 and about 10-fold improvement over ACE2-Ig-95 (Fig. 1B). A similar trend was reproduced with the pseudoviruses of the SARS-CoV-1 and a SARS-CoV-2-like coronavirus of pangolin origin (Fig. S2).

We then moved forward with ACE2-Ig-v1, ACE2-Ig-95, and ACE2-Ig-105/106 and compared them with four previously approved anti-SARS-CoV-2 monoclonal antibodies (etesevimab [LY-CoV016], bamlanivimab [LY-CoV555], casirivimab [REGN10933], and imdevimab [REGN10987]) (6–10) for their *in vitro* neutralization potencies against pseudoviruses of diverse SARS-CoV-2 variants. Luciferase reporter viruses pseudotyped with one of 15 SARS-CoV-2 Spike variants, including those of VOCs Alpha (B.1.1.7), Beta (B.1.351), Gamma (P.1), Delta (B.1.617.2), and Omicron (B.1.1.529), were tested here. We found that, while most of the RBD-mutated variants showed strong or complete resistance to at least one antibody and the B.1.1.529 variant showed strong resistance to all the four tested antibodies, all of the variants showed strong neutralization sensitivity to the three tested ACE2-Ig proteins (Fig. 1C to G; Fig. S3). More importantly, compared to the prototype WHU01 variant, most tested variants showed increased neutralization sensitivity rather than any resistance to all three ACE2-Ig constructs (Fig. 1; Fig. S3). This might be explained by increased affinity of SARS-CoV-2 variants to human ACE2 or increased accessibility of the RBDs of SARS-CoV-2 variants (5, 13, 24–26). These data demonstrate that our ACE2-Ig constructs are good drug candidates against diverse SARS-CoV-2 variants that emerged over the course of the pandemic.

**ACE2-Ig proteins with more ACE2 surface mutations neutralized SARS-CoV-2 variants less robustly.** Introduction of surface mutations to enhance ACE2-RBD interaction is a commonly used approach to engineer improved ACE2 decoys against SARS-CoV-2 (19, 20, 27–30). However, we then hypothesized that extensive ACE2 surface mutations might cause loss of neutralization potency when the heavily mutated Spike variants emerge and, more importantly, may increase the chance of eliciting antidrug antibody (ADA) immune responses when used *in vivo*; we therefore intentionally gave up this approach at a very early stage and instead improved the proteins' neutralization potency by leveraging the avidity effect of antibody-like configurations. Here, we did a head-to-head comparison of our ACE2-Ig constructs with multiple surface-mutated dimeric soluble ACE2 constructs, including one from Chan et al. (27), named here ACE2-Ig-Chan-v2.4, and four from Glasgow et al. (28), named here ACE2-Ig-Glasgow-293, ACE2-Ig-Glasgow-310, ACE2-Ig-Glasgow-311, and ACE2-Ig-Glasgow-313. Each of these surface-mutated ACE2-Ig constructs carries three to five ACE2 mutations (Fig. S4) designed to enhance ACE2 interaction with the Spike protein of the prototype SARS-CoV-2 variant (27, 28).

Here, these ACE2-Ig constructs were tested for their *in vitro* neutralization potency against 16 SARS-CoV-2 pseudoviruses: each carried a SARS-CoV-2 Spike mutant (Fig. 2; Fig. S5). We got multiple interesting findings here. First, although most of the surface-mutated ACE2-Ig constructs except for ACE2-Ig-Glasgow-293 showed neutralization potencies similar to that of ACE2-Ig-105/106 in most cases (Fig. 2A to E; Fig. S5), ACE2-

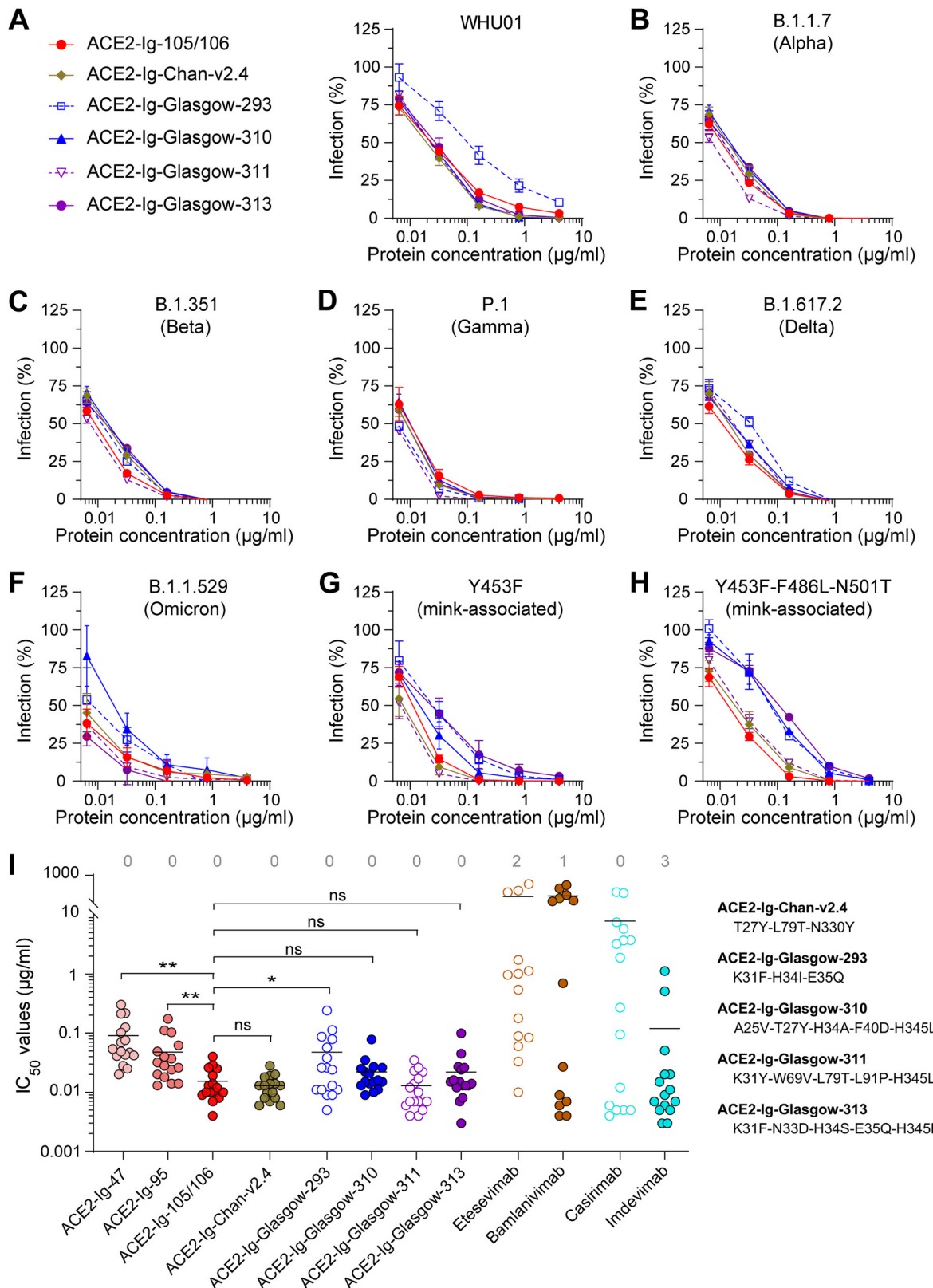

**FIG 2** Head-to-head comparison of ACE2-Ig-105/106 with five previously published ACE2-Ig constructs. (A to H) Pseudovirus neutralization experiments similar to those in Fig. 1B to G were performed to evaluate the neutralization potency and robustness of ACE2-Ig-105/106 and five surface-mutated dimeric soluble ACE2 constructs, including one from Chan et al. (27), named here ACE2-Ig-Chan-v2.4, and four from Glasgow et al. (28), named here ACE2-lg-Glasgow-293, -310, -311, and -313. The data shown are representative of three independent experiments performed by two different people with similar results, and data points represent the

Ig-Glasgow-310 and ACE2-Ig-Glasgow-313 each showed significantly weaker neutralization potency against two variants (Fig. 2F to H). More importantly, compared to the prototype variant WHU01, the mink-associated Y453F-F486L-N501T variant seemed to have partial resistance to ACE2-Ig-Glasgow-310 and ACE2-Ig-Glasgow-313 (Fig. 2H). Note that, in contrast to ACE2-Ig-105/106, which has only one very mild D30E mutation in the ACE2 region, ACE2-Ig-Glasgow-310 and ACE2-Ig-Glasgow-313 have five ACE2 surface mutations (Fig. S4). These data are clear evidence supporting our initial intention of avoiding intensive ACE2 surface mutations. Second, when the 50% inhibitory concentration ($IC_{50}$) values from studies of Fig. 1 and 2, as well as Fig. S3 and S5 in the supplemental material, were analyzed for each ACE2-Ig protein and antibody, ACE2-Ig-105/106 (average $IC_{50}$ = 37 pM), ACE2-Ig-Chan-v2.4 (average $IC_{50}$ = 59 pM), and ACE2-Ig-Glasgow-311 (average $IC_{50}$ = 58 pM) showed the best neutralization potencies and most concentrated $IC_{50}$ distributions against diverse SARS-CoV-2 variants (Fig. 2I). The more scattered $IC_{50}$ distributions observed with ACE2-Ig-Glasgow-293 and ACE2-Ig-Glasgow-313 again support our initial intention of avoiding intensive ACE2 surface mutations. These data suggest that our ACE2-Ig constructs are more likely to maintain neutralization potency against new SARS-CoV-2 variants that emerge in the future.

**ACE2-Ig-105/106 administered intranasally most efficiently lowered the lung viral load in an Ad5-hACE2-sensitized COVID-19 mouse model.** We then generated two stable CHO cell pools that express ACE2-Ig-95 and ACE2-Ig-105/106, respectively. In a 3-L scale-up culture experiment, both cell pools grew well with high cell viability during a 14-day culture period (Fig. 3A; Fig. S6A). The yields of ACE2-Ig-95 and ACE2-Ig-105/106 reached 1.6 and 0.4 g/L, respectively (Fig. 3B). Because the isoelectric points (pIs) of ACE2-Ig-95 and ACE2-Ig-105/106 are 5.65 and 5.62, respectively, purified proteins were then prepared in three different buffer formulations (F1, F2, and F3) and tested for stability under three different stress conditions (freeze-thaw, shear flow, and temperature). Although both proteins were found to be sensitive to temperature stress, ACE2-Ig-105/106 showed better stability than ACE2-Ig-95 (Fig. S6B to E). Because both proteins behaved best in buffer F3 (40 mg/mL trehalose, 0.2 mg/mL polysorbate 80, 10 mM Tris-HCl [pH 7.5]) under all three stress conditions, ACE2-Ig-95 and ACE2-Ig-105/106 were then prepared in buffer F3 for animal studies.

We first measured the plasma half-lives of the two proteins in mice. Both male and female BALB/c mice were injected intraperitoneally (i.p.) with 14 mg/kg of ACE2-Ig-95 or ACE2-Ig-105/106 protein. Blood samples were collected periodically. Quantitative enzyme-linked immunosorbent assay (ELISA) detection of the corresponding ACE2-Ig proteins from plasma samples showed that ACE2-Ig-95 and ACE2-Ig-105/106, respectively, have half-lives of 43.0 ± 1.8 h and 20.4 ± 0.3 h (Fig. 3C to F): both are markedly longer than the half-lives of recombinant ACE2 proteins without an Fc fusion (31, 32). Because ACE2-Ig-105/106 has better *in vitro* neutralization potency but shorter plasma half-life than ACE2-Ig-95, we then performed a pilot experiment using an adenovirus type 5 (Ad5)-hACE2-sensitized COVID-19 mouse model to compare i.p. administration with intranasal (i.n.) administration of the two proteins for the treatment of SARS-CoV-2 infection (33) (Fig. 3G). Each ACE2-Ig protein at 50 mg/kg of body weight was administered i.p. or i.n. to three mice per group on day 1 post-SARS-CoV-2 infection. Mice were then sacrificed on day 3 postinfection, and the lungs were harvested to measure the viral load. Among different treatments, both i.p.- and i.n.-administered ACE2-Ig-95 reduced SARS-CoV-2 viral load in the lung by ~1.5 log (Fig. 3H). ACE2-Ig-105/106 was found significantly more effective when administered via the i.n. route, and this reduced lung viral load by ~2 log (Fig. 3H).

**FIG 2** Legend (Continued)

mean ± SD from three biological replicates. (I) The $IC_{50}$ values from studies in Fig. 1B to G, Fig. 2A to H, and Fig. S3 and S5 are plotted. Each dot represents a SARS-CoV-2 variant. The numbers of SARS-CoV-2 variants resistant to 500 µg/mL of the indicated inhibitors are indicated at the top. Geometric means were calculated for neutralized isolates and are indicated with horizontal lines. The ACE2 mutations of the five surface-mutated ACE2-Ig constructs are shown to the right of the figure. One-sided two-sample *t* tests were performed for the indicated groups, and statistical significance is indicated: ns, no significance; *, $P < 0.05$; **, $P < 0.01$.

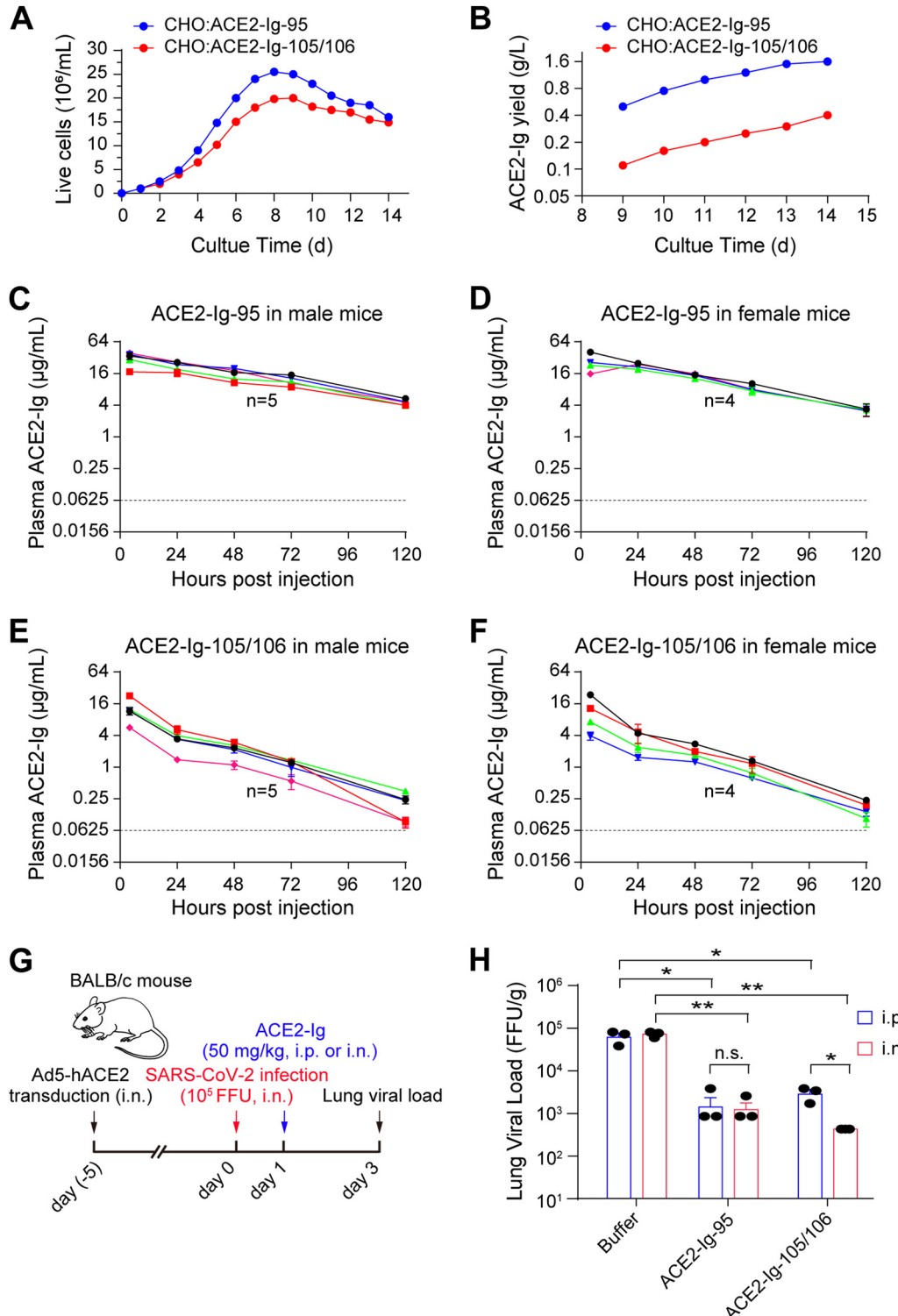

**FIG 3** Large-scale production of ACE2-Ig-95 and ACE2-Ig-105/106 and pilot experiments exploring administration routes of the proteins for *in vivo* efficacy studies. (A and B) Two stable CHO cell pools that express ACE2-Ig-95 and ACE2-Ig-105/106, respectively, were generated and tested in a 3-L scale-up culture experiment. Live cell density (A) and protein yield (B) were monitored at the indicated time points. (C to F) Male and female BALB/c mice were injected intraperitoneally (i.p.) with 14 mg/kg of ACE2-Ig-95 or ACE2-Ig-105/106 protein for protein half-life measurement. Blood samples were collected at the indicated time points, and quantitative ELISA was performed to detect the corresponding ACE2-Ig proteins from plasma samples. Each solid line represents an animal. The dashed lines represent the lower limit of detection of the quantitative ELISA. (G and H) ACE2-Ig-95 and ACE2-Ig-105/106 were tested in an Ad5-hACE2-sensitized COVID-19 mouse model (33), and administration routes (i.p. versus i.n.) for the proteins were compared. On day 1 post-SARS-CoV-2 infection, animals were i.p.

**Therapeutic use of ACE2-Ig-95 and ACE2-Ig-105/106 lowered the lung viral load and improved lung histopathology in a K18-hACE2 COVID-19 mouse model.** Based on the encouraging results from the pilot *in vivo* protection experiment, we decided to perform a more detailed evaluation of both proteins in the K18-hACE2 mouse model (22), a more commonly used and more stringent COVID animal model. As i.n. administration was shown to be comparable or superior to i.p. administration in the pilot *in vivo* protection experiment, i.n. administration was opted for in the following studies (Fig. 4A). Forty-eight K18-hACE2 mice were first intranasally infected with SARS-CoV-2 Hong Kong Isolate (hCoV-19/Hong Kong/VM20001061/2020) at 5,000 PFU. Mice were then divided into 8 groups, and treatment was initiated at 6 h postinfection. Six mice per group were treated daily for 5 consecutive days with either buffer, etesevimab at 25 mg/kg as a positive control, or ACE2-Ig-95 or ACE2-Ig-105/106 at 4, 10, or 25 mg/kg. Mice were then sacrificed on day 5 postinfection, and the lungs were harvested to measure viral load and histopathological changes. Compared to the buffer control, both proteins showed dose-dependent reduction of lung viral load, and an ~3-log reduction was observed at the 10- and 25-mg/kg doses of both proteins (Fig. 4B). ACE2-Ig-105/106 at both 10 and 25 mg/kg robustly lowered the lung viral load in all of the animals to levels close to the average lower limit of detection, indicating that the therapeutic effect of ACE2-Ig-105/106 has plateaued at the 10-mg/kg dose (Fig. 4B). As there was no significant difference in the viral loads between the animals treated with positive control (etesevimab at 25 mg/kg) and ACE2-Ig-95 or ACE2-Ig-105/106 (4 or 10 mg/kg), both ACE2-Ig proteins are considered more effective than etesevimab.

Lung histopathology analysis showed that, compared to buffer-treated mice, all of the drug-treated groups showed improvement in pulmonary lesions and lower pathological scores (Fig. 4C and D). Consistent with what we observed with the lung viral load data, mice treated with either of the ACE2-Ig proteins at 10 mg/kg showed more significant histopathological improvement than animals treated with 25 mg/kg of etesevimab, again demonstrating that both ACE2-Ig proteins are more effective than etesevimab (Fig. 4D). When each pulmonary lesion was analyzed individually, multiple lesions, including hyperplasia of alveolar type II cells, alveolar hemorrhage, congested alveolar septa, thickened alveolar walls, and interstitial inflammation were found to be significantly improved by the ACE2-Ig proteins (Fig. 4D; Fig. S7).

**ACE2-Ig-95 and ACE2-Ig-105/106 effectively protected K18-hACE2 mice from lethal SARS-CoV-2 infection.** We then performed another K18-hACE2 mouse experiment to assess the ability of ACE2-Ig-95 and ACE2-Ig-105/106 to save animals from infection-caused clinical signs and fatality. In this experiment, 64 SARS-CoV-2-infected K18-hACE2 mice were divided into eight treatment groups. Eight mice per group were treated daily for 7 consecutive days with either buffer, etesevimab at 25 mg/kg, or ACE2-Ig-95 or ACE2-Ig-105/106 at 4, 10, or 25 mg/kg (Fig. 5A). Mice were continuously monitored from day 0 through day 14 postinfection for body weight, clinical signs of SARS-CoV-2 infection, and survival. All eight mice in the buffer control group showed marked (up to 15%) body weight loss by day 7 postinfection. In contrast, the majority of the 48 ACE2-Ig-treated animals across the three treatment doses did not display infection-associated significant weight loss (Fig. 5B). Differences in body weight loss between the buffer control group and each treatment group were all significant on day 6 postinfection (one-sided two-sample *t* tests; $P < 0.05$).

For clinical score analysis, clinical signs, including piloerection, hunched posture, decreased activity, and respiration difficulty, were monitored and scored. The presence of each sign gave an animal a score of 1. The sum of the scores of an animal was defined as the animal's clinical score. A clinical score of 3 or more was considered a humane endpoint.

**FIG 3** Legend (Continued)
or i.n. treated with ACE2-Ig-95 or ACE2-Ig-105/106 at 50 mg/kg. Mice were then sacrificed on day 3 postinfection, and the SARS-CoV-2 viral load in the lung tissue was measured using a focus-forming assay. Data in panel G are presented as the mean ± SD of the lung viral load data from three animals per group. One-sided two-sample *t* tests were performed for the indicated groups, and statistical significance is indicated: n.s., no significance; *, $P < 0.05$; **, $P < 0.01$.

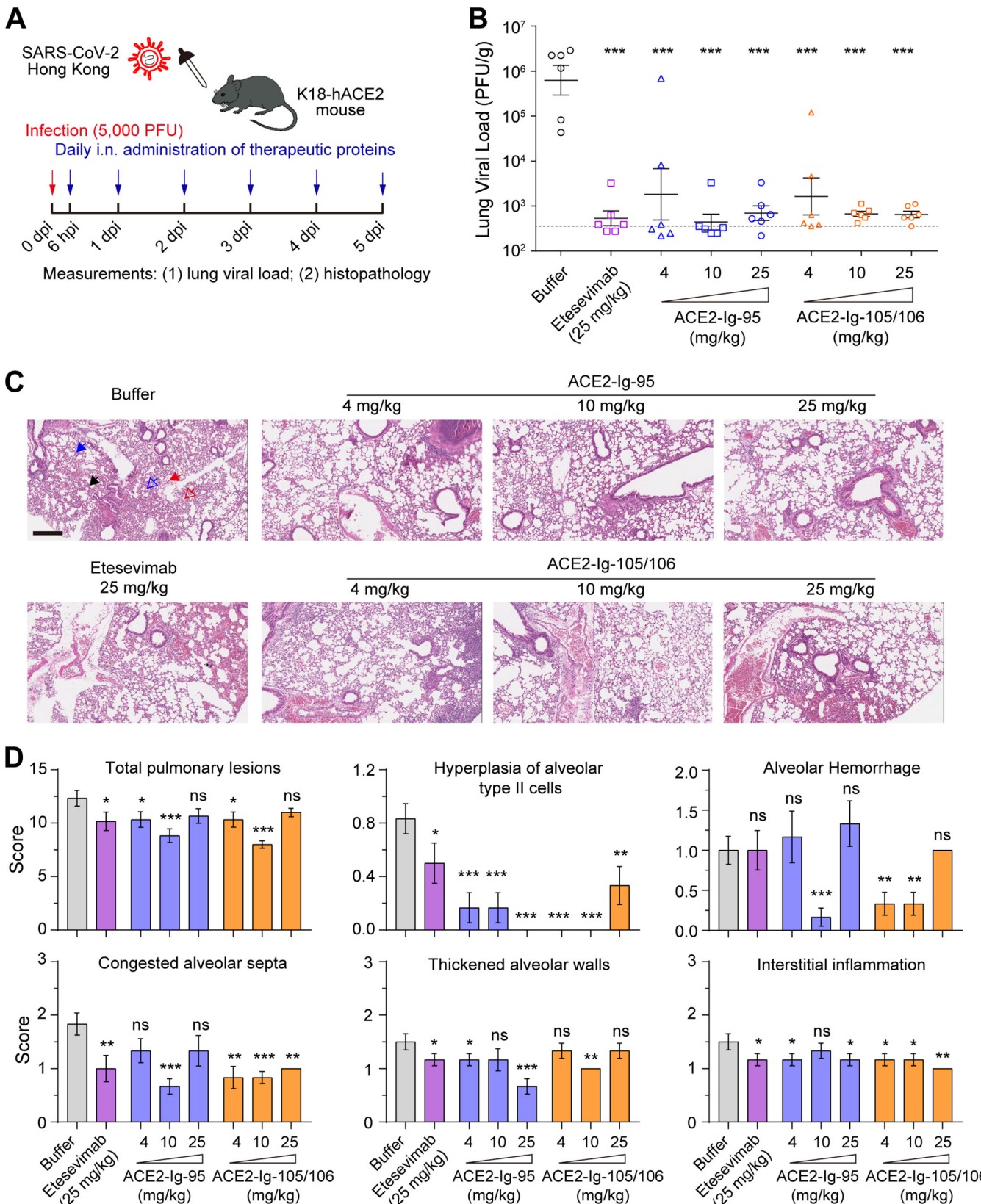

**FIG 4** ACE2-Ig-95 and ACE2-Ig-105/106 lowered the lung viral load and improved lung histopathology in SARS-CoV-2-infected K18-hACE2 mice. (A) Diagram representing the experimental design used in the following animal studies. (B to D) K18-hACE2 mice treated following the procedure in panel A ($n$ = 6 per group) were sacrificed on day 5 postinfection, and the lungs were harvested to measure the viral load (B) and histopathological changes (C and D). Each data

The clinical scores for each animal were plotted against the time of monitoring (Fig. 5C). Consistent with the body weight changes, all eight mice in the buffer control group displayed multiple SARS-CoV-2 infection-associated clinical signs and therefore high clinical scores, and most of them had clinical scores that reached the humane endpoint by day 7 postinfection. Treatment with ACE2-Ig-95 or ACE2-Ig-105/106 eliminated the development of clinical signs in most of the animals, and differences in clinical signs between the buffer control group and each treatment group were all significant on days 5 to 7 postinfection (one-sided two-sample $t$ tests; $P < 0.01$). Specifically, 2 out of 24 ACE2-Ig-95-treated mice displayed severe clinical signs that reached the humane endpoint on day 6 postinfection. Although more (5 out of 24) animals treated with ACE2-Ig-105/106 displayed clinical signs, there was likely a delay in the onset of the clinical signs, and the signs finally resolved in two of these five animals.

According to the IACUC protocol, animals with more than 20% weight loss or a clinical score of 3 or more would be euthanized and considered infection-caused fatalities. In the buffer control group, two of the mice died and the other six reached the clinical score humane endpoint by day 7 postinfection. The median survival time of this group was 6.5 days postinfection (Fig. 5D). In contrast, only 2 out of 24 animals treated with ACE2-Ig-95—one in the 4-mg/kg group and the other in the 25-mg/kg group—reached the clinical score humane endpoint on day 6 postinfection. Three out of 24 animals treated with ACE2-Ig-105/106, representing one in each dose group, died on day 8 or 9 postinfection (Fig. 5D; Fig. S8). Therefore, without treatment, the survival rate of these SARS-CoV-2 infected animals was 0%. Then, treatment with ACE2-Ig-95 and ACE2-Ig-105/106, respectively, dramatically increased the survival rates to 91.7% ($P < 0.0001$) and 87.5% ($P < 0.0001$) across the three dose groups (Fig. 5D). For the eight mice treated with 10 mg/kg of ACE2-Ig-95, the survival rate reached 100% ($P = 0.0001$) (Fig. S8B). These results demonstrate that both ACE2-Ig-95 and ACE2-Ig-105/106 are good drug candidates that can be used to protect animals from severe COVID-19.

**ACE2-Ig-95 and ACE2-Ig-105/106 are potent against diverse Omicron subvariants with extremely strong immune evasion.** During the preparation of this article, new Omicron subvariants with extremely strong immune evasion (BQ and XBB) were rapidly rising. Monoclonal antibodies capable of neutralizing the original Omicron variant were found to be largely inactive against these subvariants (12, 13). To further confirm the neutralization breadth of ACE2-Ig-95 and ACE2-Ig-105/106, especially against these highly immune-evasive BQ and XBB subvariants, we performed additional *in vitro* neutralization assays using multiple authentic SARS-CoV-2 variants, including WHU01, B.1.351 (Beta), and multiple Omicron subvariants (BA.1.1, BA.2.3, BA.5, BQ.1, and XBB.1). As expected, the antibody control (etesevimab) potently neutralized the early isolate WHU01 (Fig. 6A) but showed no neutralization activity against all the other tested variants (Fig. 6B to G). In contrast, both ACE2-Ig-95 and ACE2-Ig-105/106 showed robust and potent neutralization activities against all of the tested viruses (Fig. 6), even including the BQ.1 and XBB.1 subvariants that are resistant to most clinically used monoclonal antibodies (12, 13). More importantly, compared to the early isolate WHU01, none of the tested Omicron subvariants showed significant resistance to either ACE2-Ig-95 or ACE2-Ig-105/106 (Fig. 6H). These data further demonstrate the exceptional breadth of ACE2-Ig-95 or ACE2-Ig-105/106 and again suggest that our ACE2-Ig constructs are likely to maintain neutralization potency against new SARS-CoV-2 variants that emerge in the future.

**FIG 4** Legend (Continued)
point in panel B represents an animal, and data are presented as the mean $\pm$ SD of the lung viral load data from six animals per group. The dashed line represents the lower limit of detection of the lung viral load assay. Compared to the buffer control, all of the treatments significantly lowered the lung viral load (one-sided two-sample $t$ tests; ***, $P < 0.001$). No significance was found among treatment groups (B). The scale bar in panel C represents 300 $\mu$m. Pathological changes are indicated with different arrows: blue solid arrow, hyperplasia of alveolar type II (ATII) cells; blue open arrow, congested alveolar septa; red solid arrow, interstitial inflammation; red open arrow, alveolar hemorrhage; black arrow, thickened alveolar walls. Data points in panel D represents the mean $\pm$ SEM of the pathological scores obtained from two sections per animal and six animals per group. One-sided two-sample $t$ tests were performed between the buffer control group and each treatment group. Statistical significance is indicated: ns, no significance; *, $P < 0.05$; **, $P < 0.01$; ***, $P < 0.001$. Pathological score data for the lesions indicating that no significant difference was found between control and treatment groups are shown in Fig. S7.

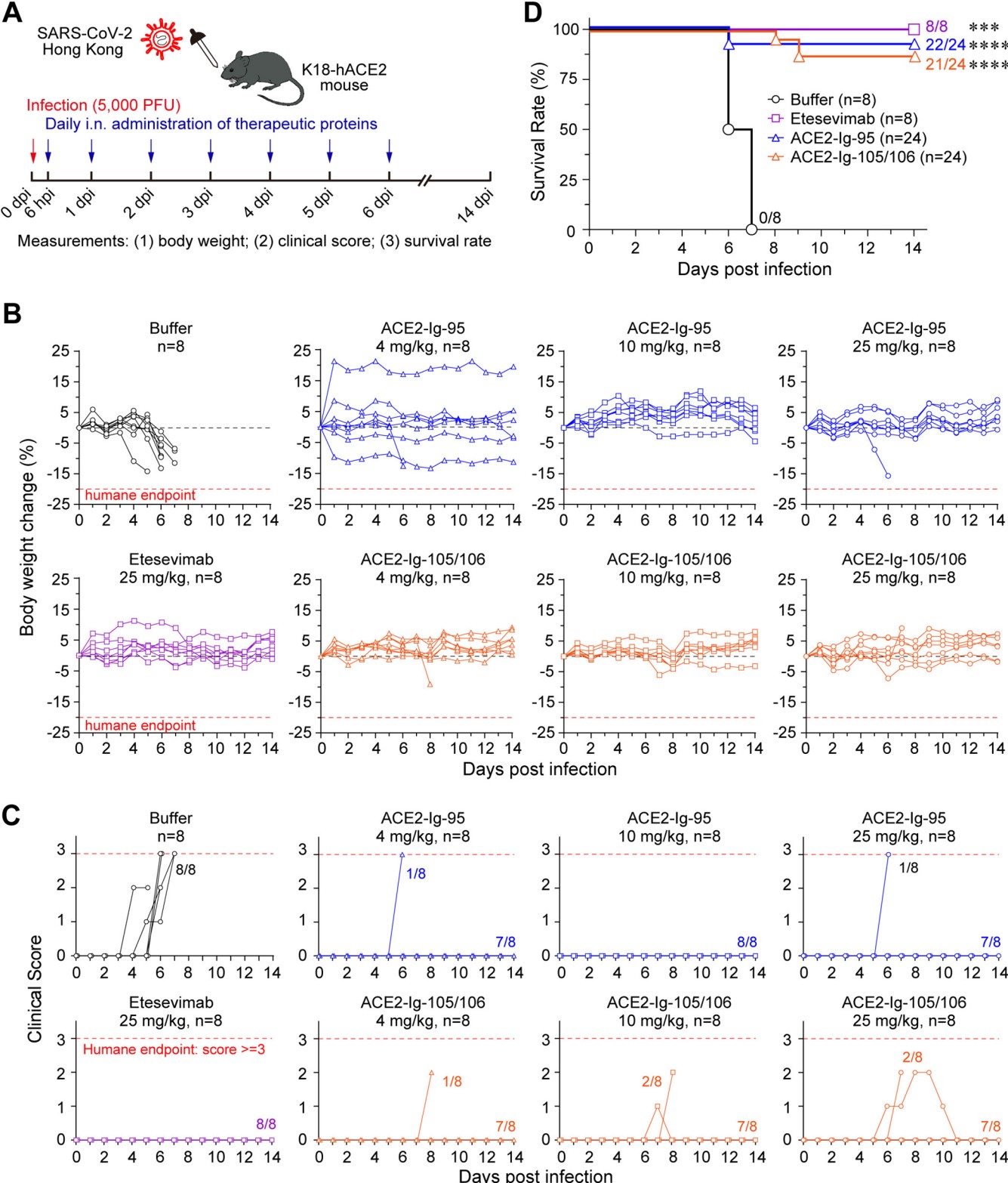

**FIG 5** ACE2-Ig-95 and ACE2-Ig-105/106 effectively protected K18-hACE2 mice from lethal SARS-CoV-2 infection. (A) Diagram representing the experimental design used in the following animal studies. (B to D) K18-hACE2 mice treated following the procedure in panel A (*n* = 8 per group) were continuously monitored from day 0 through day 14 postinfection for body weight (B), clinical signs of SARS-CoV-2 infection (C), and survival (D). Each solid line in panels B and C represents an animal, and the red dashed lines represent the humane endpoints for the studies. Differences in body weight loss between the buffer control group and each treatment group are all significant on days 6 postinfection (one-sided two-sample *t* tests; *P* < 0.05). For clinical score analysis, clinical signs, including piloerection, hunched posture, decreased activity, and respiration difficulty, were monitored and scored. The number of

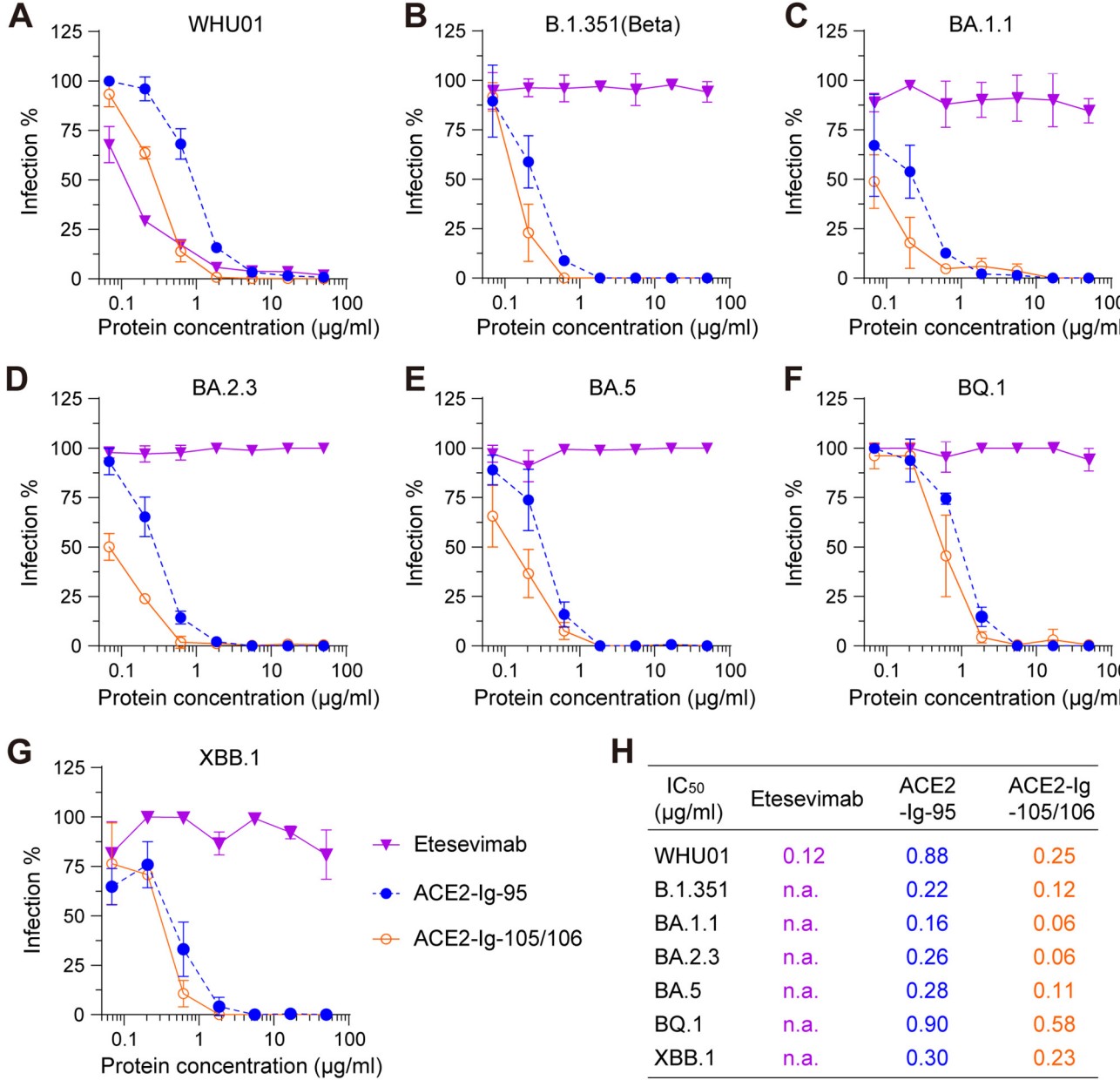

**FIG 6** ACE2-Ig-95 and ACE2-Ig-105/106 potently neutralized diverse authentic SARS-CoV-2 Omicron subvariants with extremely strong immune evasion. (A to G) ACE2-Ig-95 and ACE2-Ig-105/106 were compared with previously approved anti-SARS-CoV-2 monoclonal antibody etesevimab (LY-CoV016) for their neutralization potencies against seven authentic SARS-CoV-2 variants in Vero E6 cells. Virus infection-formed foci were detected at 24 h postinfection, and virus infection signals observed at each inhibitor concentration were divided by the signals observed at a concentration of 0 to calculate the percentage of infection values. (H) $IC_{50}$ values were calculated for the neutralization data in panels A to G. Data points in panels A to G represent the mean ± SD from three biological replicates.

## DISCUSSION

In this study, we demonstrated that, in contrast to monoclonal antibodies, ACE2-Ig-95 and ACE2-Ig-105/106 are very potent, exceptionally broad, and difficult-to-escape drug candidates against diverse SARS-CoV-2 variants. So far, most of the major SARS-

**FIG 5** Legend (Continued)
animals that displayed clinical signs in each group is indicated. Differences in clinical signs between the buffer control group and each treatment group were all significant on days 5 to 7 postinfection (one-sided two-sample *t* tests; $P < 0.01$). For survival analysis, data for animals treated with different doses of the same protein (ACE2-Ig-95 or ACE2-Ig-105/106) were pooled in the Kaplan-Meier survival curves. The number of animals that survived is indicated at the terminal point of each group. A log-rank (Mantel-Cox) test was performed to determine the statistical significance between the control and each of the treatment groups: ***, $P < 0.001$; ****, $P < 0.0001$.

CoV-2 VOCs that sequentially caused multiple waves of global transmission have been found to have ACE2-binding affinities higher than those of the original variant (5, 13, 24, 25). Structural studies have shown that Spike trimers of many SARS-CoV-2 variants have significantly higher propensity to adopt an "RBD-up" or open state than the D614G variant does (24, 26). These findings are consistent with our data that, compared to the prototype SARS-CoV-2, all of the tested variants showed either comparable or increased sensitivity to our ACE2-Ig constructs (Fig. 1, 2, and 6; see Fig. S3 in the supplemental material). These findings all suggest that ACE2-Ig is likely to be a long-term viable approach for coping with diverse circulating and emerging SARS-CoV-2 variants. In addition, a number of other coronaviruses, including SARS-CoV-1, human coronavirus NL63, and some SARS-like CoVs of bat or pangolin origin, also utilize ACE2 as an entry receptor (34–38). Recently, two close relatives of Middle East respiratory syndrome coronavirus (MERS-CoV) in bats were also found to utilize ACE2 as their functional receptor (39). Considering that coronaviruses have very broad host ranges and moderate recombination frequencies (19, 40, 41), the possibility of more coronaviruses being found to use ACE2 as their functional receptor in the future cannot be ruled out. ACE2-Ig is therefore a broadly anticoronavirus drug candidate that should be included in the toolbox for pandemic preparedness.

To develop a potent and safe ACE2-Ig for human use, four critical factors should be explored. The first critical but initially often neglected factor is the truncation of the transmembrane ACE2 protein for soluble expression. The extracellular region of ACE2 (residues 18 to 740) consists of a peptidase domain (residues 18 to 615) and a collectrin-like domain (CLD [residues 615 to 740]) (23). At the beginning of the pandemic, several groups, including us, independently explored the utility of an ACE2-Ig decoy as an anti-SARS-CoV-2 agent (19, 42–45). While most of these studies directly opted for ACE2-Ig constructs that carried the ACE2 peptidase domain but no CLD (42–45), we carefully compared a panel of CLD-free and CLD-containing ACE2-Ig constructs (19). We found that the CLD-containing ACE2-Ig constructs consistently showed ~20-fold-better neutralization potency than the CLD-free ACE2-Ig constructs (19). Since then, our ACE2-Ig studies (20, 21) (Fig. 1B; Fig. S2) and most of the ACE2-Ig studies from other groups have included the CLD in the constructs (27–29, 46). Recently, a study reported that CLD could also dramatically extend the serum half-life of their human IgG4 Fc-based ACE2-Ig constructs (46). Interestingly, with our constructs that are based on human IgG1 rather than IgG4 Fc, we did not observe this significant difference in half-lives between CLD-free and CLD-containing constructs (data not shown), suggesting the CLD's contribution to half-life might be IgG subclass dependent.

The second and most frequently investigated factor is mutations to ACE2 surface residues. Introducing mutations to ACE2 surface residues is a commonly adopted approach for engineering an improved ACE2-Ig decoy that has enhanced RBD recognition and SARS-CoV-2 neutralization activity (19, 20, 27–30). We intentionally avoided this approach at a very early stage and instead improved the proteins' neutralization potency by leveraging the avidity effect of antibody-like multivalent configurations. As a result, when we previously used *in vitro* serial viral passage to study drug-induced viral escape, no ACE2-Ig escape mutant was detected in wild-type ACE2-Ig-treated samples (21). In addition, none of the SARS-CoV-2 variants in the current study showed significantly decreased sensitivity to our ACE2-Ig constructs (Fig. 1, 2, and 6; Fig. S3). In contrast, we observed in three SARS-CoV-2 variants seemingly partial resistance to two previously published ACE2-Ig constructs, each carrying five surface mutations in the ACE2 region (Fig. 2F to H). These data clearly suggest that extensively mutating the ACE2 residues near the RBD-binding interface should be avoided or performed with extra caution. It might result in compromised neutralization breadth, not to mention the risk of eliciting ADA immune responses, which might also target endogenous ACE2 protein.

The other two important factors are ACE2 peptidase activity and Fc effector functions. Whether the peptidase activity should be retained in an ACE2-Ig product is still under debate. In this study, both ACE2-Ig-95 and ACE2-Ig-105/106 at a dose of 25 mg/

kg are counterintuitively less effective than the same proteins at 10 mg/kg in terms of histopathological lesions, clinical score, and survival rate (Fig. 4 and 5). This might be associated with the peptidase activity of ACE2 in both proteins. Although some previous non-COVID studies in animals as well as in humans (47–49) support a potentially beneficial and non-antiviral role of ACE2 peptidase activity in COVID-19 treatment, more detailed *in vivo* studies to dissect the contribution of ACE2 peptidase activity to the treatment are needed in the future. Fc effector functions are normally beneficial for antibody-like antiviral biologics (50). Indeed, a recent study reported that Fc effector functions could enhance ACE2-Ig's therapeutic activity in a COVID-19 mouse model (29). However, because human ACE2 is an endogenous protein with broad substrate specificity (51, 52), keeping or enhancing ACE2-Ig's Fc effector functions might cause off-target cell killing. Therefore, more carefully designed studies to investigate this aspect should be warranted in the future. An ACE2-Ig product that keeps Fc effect functions has moved into the clinical stage (53) (ClinicalTrials registration no. NCT05116865 and NCT05659602). Safety data from these trials should be informative.

In terms of the four critical factors, both ACE2-Ig-95 and ACE2-Ig-105/106 carry a CLD-containing, barely mutated, and peptidase-active ACE2 ectodomain and an effector function-competent IgG1 Fc. They neutralized diverse SARS-CoV-2 variants with robust potencies (Fig. 1, 2, and 6) and markedly lowered the lung viral loads in two different COVID-19 mouse models (Fig. 3 and 4). In the more stringent K18-hACE2 mouse model, both proteins at 4-mg/kg daily doses effectively prevented the emergence of clinical signs and greatly increased survival rate of the animals (Fig. 5). These data demonstrate that ACE2-Ig-95 and ACE2-Ig-105/106 are broadly effective and hard-to-escape anti-SARS-CoV-2 drug candidates that can be used to protect animals from severe COVID-19. Besides efficacy, producibility and stability are critical determinants for the developability of a biologic drug candidate. Three-liter scale-up culture tests showed that both ACE2-Ig-95 and ACE2-Ig-105/106 stable cells had a yield at the level of grams per liter (Fig. 3B), demonstrating that both proteins as biologic drug candidates are developable. However, the yield of ACE2-Ig-105/106 is 4-fold lower than that of ACE2-Ig-95, possibly because the more complicated tetramer configuration of ACE2-Ig-105/106 could result in higher chance of failure to produce fully assembled ACE2-Ig-105/106 tetramers and more protein-folding stress in the cells (54), which is consistent with the observation that the ACE2-Ig-105/106 stable cell pool has a lower growth rate than the ACE2-Ig-95 stable cell pool (Fig. 3A). Moreover, stability tests showed that a significant fraction of both proteins, especially ACE2-Ig-95, aggregated under a temperature stress condition (Fig. S6). Therefore, more in-depth formulation or engineering studies should be performed in the future to address the stability of these proteins.

## MATERIALS AND METHODS

**Cells.** 293T cells and HeLa cells were kindly provided by Stem Cell Bank, Chinese Academy of Sciences, confirmed mycoplasma free by the provider, and maintained in Dulbecco's modified Eagle's medium (DMEM) (Life Technologies) at 37°C in a 5% $CO_2$-humidified incubator. Growth medium was supplemented with 2 mM Glutamax-I (Gibco; catalog no. 35050061), 100 $\mu$M nonessential amino acids (Gibco; catalog no. 11140050), 100 U/mL penicillin and 100 $\mu$g/mL streptomycin (Gibco; catalog no. 15140122), and 10% heat-inactivated fetal bovine serum (FBS) (Gibco; catalog no. 10099141C). HeLa cell-based stable cells expressing human ACE2 were maintained under the same culture condition as HeLa cells, except that 3 $\mu$g/mL of puromycin was added to the growth medium. 293F cells for recombinant protein production were generously provided by Yu J. Cao (School of Chemical Biology and Biotechnology, Peking University Shenzhen Graduate School) and maintained in SMM 293-TII serum-free medium (Sino Biological; catalog no. M293TII) at 37°C in 8% $CO_2$ in a shaker incubator at 125 rpm.

**Plasmids.** A DNA fragment encoding spike protein of SARS-CoV-2 WHU01 (GenBank accession no. MN988668.1) was synthesized by the Beijing Genomic Institute (BGI [China]) and then cloned into pCDNA3.1(+) plasmid between EcoRI and XhoI restriction sites. Plasmids encoding SARS-CoV-2 Spike variants were generated according to the In-Fusion Cloning protocol. To facilitate SARS-CoV-2 pseudovirus production, Spike sequences for WHU01 and all of the variants investigated in this study all contain a deletion (ΔPRRA) or GSAS substitution at the PRRA furin cleavage site. Our previous study showed that the ΔPRRA mutation does not affect SARS-CoV-2 cross-species receptor usage or neutralization sensitivity (20). The retroviral reporter plasmids encoding a *Gaussia* luciferase reporter gene were constructed by cloning the reporter genes into the pQCXIP plasmid (Clontech). Plasmids encoding soluble ACE2 variants fused with human IgG1 Fc were described in our previous study (20). DNA fragments encoding

heavy and light chains of anti-SARS-CoV-2 antibodies were synthesized by Sangon Biotech (Shanghai, China) and then cloned into the pCAGGS plasmid.

**Production and Purification of ACE2-Ig protein and SARS-CoV-2 antibodies by transient transfection.** 293F cells at a density of $6 \times 10^5$ cells/mL were seeded into 100 mL SMM 293-TII serum-free medium (Sino Biological; catalog no. M293TII) 1 day before transfection. Cells were then transfected with 100 $\mu$g plasmid in complex with 250 $\mu$g polyethyleneimine (PEI) MAX 40000 (Polysciences; catalog no. 24765-1). Cell culture supernatants were collected at 48 to 72 h posttransfection. Human IgG1 Fc-containing proteins were purified using protein A Sepharose CL-4B (GE Healthcare; catalog no. 17-0780-01), eluted with 0.1 M citric acid at pH 4.5, and neutralized with 1 M Tris-HCl at pH 9.0. Buffers were then changed to phosphate-buffered saline (PBS), and proteins were concentrated by 30-kDa-cutoff Amicon Ultra-15 centrifugal filter units (Millipore; catalog no. UFC903096).

**Production of reporter retroviruses pseudotyped with SARS-CoV-2 spike variants.** Murine leukemia virus (MLV) retroviral vector-based SARS-CoV-2 Spike pseudotypes were produced according to our previous study (20), with minor changes. In brief, 293T cells were seeded at a 30% density in a 150-mm dish 12 to 15 h before transfection. Cells were then transfected with 67.5 $\mu$g of polyethyleneimine PEI MAX 40000 (Polysciences, Inc.; catalog no. 24765-1) in complex with 3.15 $\mu$g of plasmid encoding a Spike variant, 15.75 $\mu$g of plasmid encoding MLV Gag and Pol proteins, and 15.75 $\mu$g of a pQCXIP-based luciferase reporter plasmid. Eight hours after transfection, cell culture medium was refreshed and changed to growth medium containing 2% FBS (Gibco; catalog no. 10099141C) and 25 mM HEPES (Gibco; catalog no. 15630080). Cell culture supernatants were collected 36 to 48 h posttransfection, spun down at $3,000 \times g$ for 10 min, and filtered through 0.45-$\mu$m-pore filter units to remove cell debris. SARS-CoV-2 spike-pseudotyped viruses were then concentrated 10-fold at $2,000 \times g$ using 100-kDa-cutoff Amicon Ultra-15 centrifugal filter units (Millipore; catalog no. UFC910024).

**Pseudovirus titration.** Pseudovirus titers were determined using a reverse transcriptase activity assay. Reverse transcriptase-containing pseudoviral particles and a recombinant reverse transcriptase standard of known concentrations (TaKaRa; catalog no. RR047A) were 10-fold diluted with nuclease-free water (Invitrogen; catalog no. 10977015) and lysed with 2-fold-concentrated lysis buffer (0.25% Triton X-100, 50 mM KCl, 100 mM Tris-HCl [pH 7.4], 40% glycerol, 1/50 volume of RNase inhibitor [NEB; catalog no. M0314S] at room temperature for 10 min). Reverse transcription was performed according to the manufacturer's protocol (TaKaRa; catalog no. RR047A) using 1 $\mu$L of the lysate as reverse transcriptase and TRIzol reagent-isolated 293T total RNA as the template. Reverse transcription products were then subjected to quantitative PCR (qPCR) with a commercial kit (TaKaRa; catalog no. RR820Q) to amplify GAPDH glyceraldehyde-3-phosphate dehydrogenase (forward primer, 5′-CCACTCCTCCACCTTTGAC-3′; reverse primer, 5′-ACCCTGTTGCTGTAGCCA-3′) in Applied Biosystems QuantStudio 5. A standard curve was generated based on qPCR threshold cycle ($C_T$) values obtained with serially diluted recombinant reverse transcriptase standard.

**SARS-CoV-2 pseudovirus neutralization assay.** Pseudovirus neutralization experiments were performed following our previous study (20), with minor changes. In brief, SARS-CoV-2 spike variant pseudotyped luciferase reporter viruses equivalent to $8 \times 10^{10}$ U reverse transcriptase were prediluted in DMEM (with 2% FBS, heat inactivated) containing titrated amounts of an ACE2-Ig construct or an anti-SARS-CoV-2 antibody. Virus-inhibitor mixtures were incubated at 37°C for 30 min and then added to HeLa-hACE2 cells in 96-well plates and incubated overnight at 37°C. Virus-inhibitor-containing supernatant was then removed, changed to 150 $\mu$L of fresh DMEM (with 2% FBS), and incubated at 37°C. Cell culture supernatants were collected for a *Gaussia* luciferase assay at 48 h postinfection.

***Gaussia* luciferase luminescence flash assay.** To measure *Gaussia* luciferase expression, 20 $\mu$L of cell culture supernatant of each sample and 100 $\mu$L of assay buffer containing 4 $\mu$M coelenterazine native (Biosynth Carbosynth; catalog no. C-7001) was added to one well of a 96-well black opaque assay plate (Corning; catalog no. 3915) and measured with Centro LB 960 microplate luminometer (Berthold Technologies) for 0.1 s/well.

**Stable CHO cell generation and 3-L scale-up production of ACE2-Ig-95 and ACE2-Ig-105/106.** Two CHOZN CHO K1-based stable cell pools stably expressing ACE2-Ig-95 and ACE2-Ig-105/106, respectively, were generated and tested for 3-L scale-up production and stress condition stability by Canton Biologics (Guangzhou, China). In brief, CHOZN CHO K1 cells were thawed and maintained in EX-CELL CD CHO fusion medium (Sigma; catalog no. 14365C) containing 4 mM L-glutamine at 37°C in 5% $CO_2$ at 85% humidity with 140-rpm agitation. Cells were then transfected with ACE2-Ig-95 or ACE2-Ig-105/106 plasmid using PEI 25K (Polysciences; catalog no. 23966-1) and selected using methionine sulfoximine (MSX)-containing EX-CELL CD CHO fusion medium (Sigma; catalog no. 14365C). Cells passed the MSX selection cycles were then subjected to a pilot production experiment. Cells were maintained in a 280-mL culture for 14 days in EX-CELL Advanced CHO fed-batch medium (Sigma; catalog no. 14366C), to which Cell Boost 7a (HyClone; catalog no. SH31119.01) and Cell Boost 7b (HyClone; catalog no. SH31120.01) were then added. Cell viability, live cell density, and protein expression were monitored on a daily basis from day 3 through day 14 of the culture period. Protein production-validated cells were then subjected to a scale-up production experiment in 3-L bioreactors (Applikon my-Control). Cells were initially diluted to $0.5 \times 10^6$ cells/mL in 1.2 L EX-CELL Advanced CHO fed-batch medium (Sigma; catalog no. 14366C). Starting from day 3, cells were added daily with glucose to 8 g/L, a 3% volume of Cell Boost 7a (HyClone; catalog no. SH31119.01), and a 0.3% volume of Cell Boost 7b (HyClone; catalog no. SH31120.01) until day 14. Cells were maintained at 37°C in 40% dissolved oxygen and stirred at 300 rpm/320 rpm with gas flow rates of 33 mL/min and 12 mL/min (0.01 vol/vol/min). Cell viability, live cell density, and protein expression (protein A high-performance liquid chromatography [HPLC]) were monitored on daily basis through the 14-day culture period. ACE2-Ig-95 in cell culture supernatant was first captured using MabSelect SuRe affinity column (purity,

83.73%; recovery rate, 102%) and then purified using a UniHR phenyl 30L hydrophobic interaction chromatography column (purity, ~95%; recovery rate, 32%). ACE2-Ig-105/106 in cell culture supernatant was first captured using MabSelect SuRe affinity column (purity, 85%; recovery rate, 85%) and then purified using a Diamond Q Mustang anion-exchange column (purity, ~94%; recovery rate, 70%).

**ACE2-Ig-95 and ACE2-Ig-105/106 stability tests under stress conditions.** Because the isoelectric points (pIs) of ACE2-Ig-95 and ACE2-Ig-105/106 are 5.65 and 5.62, respectively, the proteins were prepared at a 10-mg/mL concentration in three different buffers (F1, F2, and F3). All three buffers contain 40 mg/mL trehalose and 0.2 mg/mL polysorbate 80. In addition, buffers F1 (pH 6.5) and F2 (pH 7.0) have 10 mM histidine. Buffer F3 (pH 7.5) has 10 mM Tris-HCl. Proteins in these different buffers were then assessed for their stability under the following three stress conditions: freeze-thaw stress (five cycles of freezing at $-80°C$ and thawing at room temperature), shear stress (agitation at 300 rpm at 37°C for 1 week), and temperature stress (incubation at 40°C for 2 weeks). Protein samples were then subjected to size exclusion chromatography (SEC) analysis using a TSKgel G3000SW$_{XL}$ column to quantify the fractions of high molecular weight, main peak, and low molecular weight, respectively.

**Focus-forming assay for SARS-CoV-2 quantification.** For the focus-forming assay (FFA), all SARS-CoV-2 live virus infection experiments were performed in a biosafety level 3 (BSL3) laboratory. Vero E6 cells were seeded onto 96-well plates overnight and grown into confluent monolayers. Fifty microliters of 10-fold-diluted SARS-CoV-2 stock or supernatant of lung homogenate was added into a 96-well plate and adsorbed at 37°C for 1 h with agitation every 10 min. Then, the virus or supernatant of lung homogenate was removed and covered with 100 $\mu$L minimum essential medium (MEM) containing 1.2% carboxymethylcellulose (CMC). Twenty-four hours postinfection, the overlay was discarded and the cell monolayer was fixed with 4% paraformaldehyde solution for 2 h at room temperature. After being permeabilized with 0.2% Triton X-100 for 20 min at room temperature, the plates were sequentially stained with cross-reactive rabbit anti-SARS-CoV-N IgG (Sino Biological, Inc.) as the primary antibody and horseradish peroxidase (HRP)-conjugated goat anti-rabbit IgG (H+L) (Jackson ImmunoResearch) as the secondary antibody at 37°C for 1 h. The reactions were developed with KPL TrueBlue peroxidase substrates. The numbers of SARS-CoV-2 foci were calculated using CTL ImmunoSpot S6 Ultra reader (Cellular Technology, Ltd.), and titers of the virus were expressed as focus-forming units (FFU) per milliliter.

**SARS-CoV-2 live virus *in vitro* neutralization assay.** The SARS-CoV-2 variants, including WHU01, Beta (B.1.351), and Omicron subvariants (BA.1.1, BA.2.3, BA.5, XBB.1, and BQ.1), were preserved in the Guangzhou Customs District Technology Center BSL3 laboratory. The SARS-CoV-2 variants BA.5, XBB.1 and BQ.1 were provided by the Guangdong Provincial Center for Disease Control and Prevention, China. Experiments related to authentic SARS-CoV-2 neutralization were conducted in the BSL3 laboratory of the Guangzhou Customs District Technology Center. Briefly, prediluted antibody or ACE2-Ig proteins were mixed with 100 FFU of SARS-CoV-2 and coincubated at 37°C for 1 h. Then, 50 $\mu$L of mixture was transferred to a 96-well plate preseeded with Vero E6 cells. After being maintained at 37°C for 1 h, the cells were covered with MEM containing 1.6% CMC and cultured for 24 h. Cells were then fixed with 4% paraformaldehyde solution, permeabilized with 0.2% Triton X-100, and incubated with anti SARS-CoV/SARS-CoV-2 nucleocapsid rabbit polyclonal antibody (PAb) (Sino Biological; no 40143-T62) and HRP-conjugated goat anti-rabbit IgG (H+L) (Jackson ImmunoResearch; no. 109-035-088). Virus infection foci were stained by KPL TrueBlue peroxidase substrate and read by a CTL ImmunoSpot S6 Ultra reader (Cellular Technology, Ltd.).

**SARS-CoV-2 live virus studies in Ad5-hACE2-sensitized mice.** Ad5-hACE2-sensitized mice were used to evaluate *in vivo* efficacy of ACE2-Ig-95 and ACE2-Ig-105/106 following our previous study (33). These experiments were conducted in a BSL3 laboratory under protocols approved by the Institutional Animal Care and Use Committee (IACUC) in the Guangzhou Customs District Technology Center. Briefly, 6-week-old female BALB/c mice (~15 g) were first intranasally transduced with 2.5 × 10$^8$ FFU of Ad5-hACE2. Five days later, animals were then intranasally challenged with 1 × 10$^5$ FFU of SARS-CoV-2. The SARS-CoV-2 strains used in this research were isolated from COVID-19 patients in Guangzhou and in Washington state (accession no. MT123290 and MN985325.1) and passaged on Vero E6 and Calu-3 2B4 cells. On day 1 post-SARS-CoV-2 infection, animals were intraperitoneally injected with an ACE2-Ig protein at 50 mg/kg. Animals were then sacrificed on day 3 post-SARS-CoV-2 infection, and the lungs were collected in PBS and homogenized. Titers of SARS-CoV-2 in clarified supernatants were determined using the FFA in Vero E6 cells and expressed as FFU per gram of tissue.

**SARS-CoV-2 live virus studies in K18-hACE2 mice.** Six- to -8-week-old specific-pathogen-free female B6.Cg-Tg(K18-ACE2)2Prlmn/J transgenic mice (called K18-hACE2 mice here) were purchased from the Jackson Laboratory. All K18-hACE2 mouse experiments were performed in a BSL3 laboratory under approved IACUC protocols by a Wuxi AppTech sponsored research institution. Animals were housed in individually ventilated cages and randomly assigned to different treatment groups. Each treatment group had six or eight mice. On day 0, animals were intranasally infected with 5,000 PFU SARS-CoV-2 Hong Kong isolate (Hong Kong/VM20001061/2020; ATCC) in a 50-$\mu$L volume.

For lung viral load and histopathological analysis, 48 SARS-CoV-2-infected K18-hACE2 mice were divided into 8 groups, and treatment was initiated at 6 h postinfection. Six mice per group were treated daily for 5 consecutive days with either buffer, etesevimab at 25 mg/kg as a positive control, or ACE2-Ig-95 or ACE2-Ig-105/106 at 4, 10, or 25 mg/kg. Mice were then sacrificed on day 5 postinfection. The left lungs of the animals were harvested and fixed in 10% formalin for histopathological analysis. The right lungs of the animals were collected, weighed, and stored in Eagle's minimal essential medium (EMEM) containing 1% FBS at $-80°C$ for lung viral load analysis.

For survival analysis, 64 SARS-CoV-2-infected K18-hACE2 mice were divided into eight treatment groups and treatment was initiated at 6 h postinfection. Eight mice per group were treated daily for 7

consecutive days with either buffer, etesevimab at 25 mg/kg, or ACE2-Ig-95 or ACE2-Ig-105/106 at 4, 10, or 25 mg/kg. Mice were continuously monitored from day 0 through day 14 postinfection for body weight, clinical signs of SARS-CoV-2 infection, and survival. For clinical sign monitoring, piloerection, hunched posture, decreased activity, and respiration difficulty were monitored and scored. The presence of each sign gave an animal a score of 1. The sum of the scores of an animal was defined as the animal's clinical score. According to the IACUC protocol, the humane endpoint is defined as body weight loss of 20% or more, a clinical score of 3 or more, or the agonal state. An animal that has reached the humane endpoint will be euthanized.

**Lung viral load measurement using plaque assay.** Vero E6 cells were seeded into 6-well plates to a $7.5 \times 10^5$ cell/mL density before infection. The right-lung samples collected in EMEM with 1% FBS were homogenized in a tissue homogenizer. Samples were then centrifuged, and the supernatant was 10-fold serially diluted and used to infect Vero E6 cells for 1 h at 37°C with shaking at 15-min intervals. Cell culture supernatant was then removed, and 1% agarose in EMEM supplemented with 20% FBS was added to the cells and incubated for 3 days at 37°C. Agarose was then carefully removed, and the cells were first fixed with 95% ethanol for 15 min. After a brief wash with PBS, the cells were then fixed and stained for 15 min in 10% formalin containing 1% crystal violet. SARS-CoV-2 infection-caused plaques were counted, and viral titers were finally calculated and converted to PFU per gram of lung tissue.

**Lung histopathological analysis.** The left lungs fixed in 10% formalin were paraffin embedded and sectioned. Sections were stained with hematoxylin and eosin for histopathological analysis. Photomicrographs taken with the Leica Aperio AT2 digital pathology scanning system (Leica Biosystems, Inc.) were subjected to semiquantitative histopathological analysis. The photomicrographs were analyzed for the presence of the following alveolar region lesions (pulmonary edema, alveolar hemorrhage, thickened alveolar walls, alveolar inflammation, necrosis, hyaline membrane, thrombus, hyperplasia of alveolar type II cells, and alveolar space protein fragments) and mesenchymal region lesions (interstitial inflammation, congested alveolar septa, perivascular edema, and perivascular hemorrhage). The presence of each lesion gave a photomicrograph a grade score between 0 to 5 according to the severity of the lesion. A score of 0 means no lesion, 1 means minimal, 2 means slight, 3 means moderate, 4 means marked, and 5 means severe.

**Data collection and analysis.** MikroWin 2000 software (Berthold Technologies) was used to collect luciferase assay data. The Leica Aperio AT2 digital pathology scanning system (Leica Biosystems, Inc.) was used to collect the photomicrographs for the lung histopathological analysis. GraphPad Prism 9.4 software was used for figure preparation and statistical analyses.

**Statistical analysis.** All of the *in vitro* experiments were independently performed two or three times, and data are expressed as the mean value ± standard deviation (SD) or standard error of the mean (SEM). Statistical analyses were performed using a two-sample *t* test ($IC_{50}$, lung viral load, and histopathology) or log rank (Mantel-Cox) test (survival) when applicable. Differences were considered significant at $P < 0.05$. The values for *n*, *P*, and the specific statistical test performed for each experiment are included in the figure legend and main text.

**Data availability.** All data are available in the main text or the supplemental materials. This study did not generate unique data sets or code. Our in-house research resources, including methods, plasmids, and protocols, are available upon reasonable request to qualified academic investigators for noncommercial research purposes. All reagents developed in house, including vector plasmids and detailed methods, will be made available upon written request.

## SUPPLEMENTAL MATERIAL

Supplemental material is available online only.

**SUPPLEMENTAL FILE 1**, PDF file, 2.1 MB.

## ACKNOWLEDGMENTS

Funding for this study was provided by Shenzhen Bay Laboratory Major Program grant S201101001-2 (G.Z. and Y.L.) and Shenzhen Bay Laboratory Key COVID-19 Program grant S211410002 (G.Z. and Y.L.).

Conceptualization, G.Z.; Methodology, W.Y., Y.L., H.W., X.T., Y.W., C.L., D.C., H.L., and G.Z.; Investigation, M.L., W.Y., D.M., Z.Z., and Y.L.; Supervision, Y.L., Y.Y., J.Z., and G.Z.; Writing – Original Draft, G.Z., M.L., and H.W.; Writing – Review & Editing, M.L., W.Y., Y.L., D.M., Z.Z., H.W., X.T., Y.W., C.L., D.C., H.L., Y.Y., J.Z., and G.Z.

We declare no conflict of interest.

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
