## [Reviewer comments · Microbiology Spectrum]

Microbiology Spectrum

Broadly effective ACE2 decoy proteins protect mice from lethal SARS-CoV-2 infection

Mengjia Lu, Weitong Yao, Yujun Li, Danting Ma, zhaoyong zhang, Haimin Wang, Xiaojuan Tang, Yanqun Wang, Chao Li, Dechun Cheng, Hua Lin, Yandong Yin, Jincun Zhao, and Guocai Zhong

Corresponding Author(s): Guocai Zhong, University of Massachusetts Chan Medical School

Review Timeline:

Submission Date:	March 14, 2023
Editorial Decision:	April 19, 2023
Revision Received:	May 31, 2023
Accepted:	June 6, 2023

Editor: Tao Deng

Reviewer(s): Disclosure of reviewer identity is with reference to reviewer comments included in decision letter(s). The following individuals involved in review of your submission have agreed to reveal their identity: Ruangang Pan (Reviewer #1)

Transaction Report:

DOI: <https://doi.org/10.1128/spectrum.01100-23>

April 19, 2023

Dr. Guocai Zhong
University of Massachusetts Chan Medical School
Biochemistry and Molecular Biotechnology
364 Plantation Street LRB-815
Worcester, MA 01605

Re: Spectrum01100-23 (Broadly effective ACE2 decoy proteins protect mice from lethal SARS-CoV-2 infection)

Dear Dr. Guocai Zhong:

Link Not Available

Sincerely,

Tao Deng

Journals Department
Reviewer comments:

Reviewer #1 (Comments for the Author):

The emergence and epidemic of SARS-CoV-2 variants continue to present challenges to human health. The development of safe, efficacious prophylactic and therapeutic drugs for SARS-CoV-2 variant infection is urgently needed. In the submitted manuscript, Lu et al. report two potential therapeutic drug candidates for SARS-CoV-2 variant infection. The authors designed and characterized two soluble SARS-CoV-2 receptor decoy proteins (antibody-like constructs), ACE2-Ig-95 and ACE2-Ig-105/106. The data show that both proteins exhibit potent and robust in vitro neutralization activities and broadly block diverse SARS-CoV-2 variant infections. Further studies demonstrate that the administration of ACE2-Ig-95 or ACE2-Ig-105/106 significantly reduces lung viral load and pulmonary histopathological lesions, and protects animals against lethal SARS-CoV-2 infection using a K18-hACE2 COVID-19 mouse model. Both ACE2-Ig-95 and ACE2-Ig-105/106 achieve high expression levels in CHO K1-based stable cell lines and exhibit good stability. These results identify the two engineered proteins as attractive drug candidates for SARS-CoV-2 infection. The experimental design is well-reasoned, the data are solid, and the conclusion is convincing.

However, the following concerns need to be addressed:

- 1) In CHO K1-based stable cell lines, the yield of ACE2-Ig-95 and ACE2-Ig-105/106 are 1.6 g/L and 0.4 g/L, respectively. Why is there a tremendous difference in expression levels? The authors need discuss this in the Discussion section.
- 2) In BALB/c mice, ACE2-Ig-95 has a longer half-life compared to ACE2-Ig-105/106 (43.0 {plus minus} 1.8 vs 20.4 {plus minus} 0.3 h) (lines 225-226; Figure 3C-3F); however, similar efficacy in reducing pulmonary viral load can be observed, especially via the i.n. route (Figure 3H).
- 3) Compared to the positive control (Etesevimab), ACE2-Ig-95 and ACE2-Ig-105/106 exhibit stronger neutralization activities (Figure 1 and Figure 2); however, the animals treated with the positive control (Etesevimab) seem to display better therapeutic effects than those treated with ACE2-Ig-95 or ACE2-Ig-105/106 at the same dosage (25 mg/kg) (Figure 4 and Figure 5). Additionally, among the treatment dosages, the 10 mg/kg dose of ACE2-Ig-95 or ACE2-Ig-105/106 provides more effective protection against SARS-CoV-2 infection, including survival rate, pulmonary viral load, and histopathological lesions, et al. (Figure 4 and Figure 5). How can this observation be explained?
- 4) It would be beneficial for the authors to review the manuscript for consistency in grammar and word usage. For example, a uniform description for the unit of protein concentration should be used throughout the manuscript, either "gram/liter" (line 50) or "g/L" (lines 213, 395, 499).

Reviewer #2 (Comments for the Author):

This study reports the development of two human ACE2 decoy proteins ACE2-Ig-95 and ACE2-Ig-105/106 and demonstrated these two proteins have a potent in vitro neutralization activity against tested SARS-CoV-2 variants. In lethal human ACE2 transgenic mice, the proteins could significantly decrease the viral load in lung tissue and alleviate the severity and mortality. In comparison with previously reported ACE2-Ig proteins with residue mutations, the study showed that the ACE2 mutations may lose the neutralization potency against the SARS-CoV-2. Furthermore, the study demonstrated the high production capacity in the laboratory. The study was well designed and the data is solid, which provide alternative route for SARS-CoV-2 treatment.

Minor comments:

1. In addition to the SARS-CoV-2 variants used in in vitro study, authors please indicate clearly which strains and how many dose used in different animal challenge studies.
2. Authors should discuss why the 10 mg/kg is more potent (Figure 4)?

Staff Comments:

Preparing Revision Guidelines

Please return the manuscript within 60 days; if you cannot complete the modification within this time period, please contact me. If you do not wish to modify the manuscript and prefer to submit it to another journal, please notify me of your decision immediately so that the manuscript may be formally withdrawn from consideration by Microbiology Spectrum.

Responses to Reviewer #1 (reviewer comments in black; author responses in blue)

Reviewer #1 (Comments for the Author):

The emergence and epidemic of SARS-CoV-2 variants continue to present challenges to human health. The development of safe, efficacious prophylactic and therapeutic drugs for SARS-CoV-2 variant infection is urgently needed. In the submitted manuscript, Lu et al. report two potential therapeutic drug candidates for SARS-CoV-2 variant infection. The authors designed and characterized two soluble SARS-CoV-2 receptor decoy proteins (antibody-like constructs), ACE2-Ig-95 and ACE2-Ig-105/106. The data show that both proteins exhibit potent and robust *in vitro* neutralization activities and broadly block diverse SARS-CoV-2 variant infections. Further studies demonstrate that the administration of ACE2-Ig-95 or ACE2-Ig-105/106 significantly reduces lung viral load and pulmonary histopathological lesions, and protects animals against lethal SARS-CoV-2 infection using a K18-hACE2 COVID-19 mouse model. Both ACE2-Ig-95 and ACE2-Ig-105/106 achieve high expression levels in CHO K1-based stable cell lines and exhibit good stability. These results identify the two engineered proteins as attractive drug candidates for SARS-CoV-2 infection. The experimental design is well-reasoned, the data are solid, and the conclusion is convincing.

We appreciate the reviewer's careful evaluation and positive comments on our study.

However, the following concerns need to be addressed:

1) In CHO K1-based stable cell lines, the yield of ACE2-Ig-95 and ACE2-Ig-105/106 are 1.6 g/L and 0.4 g/L, respectively. Why is there a tremendous difference in expression levels? The authors need discuss this in the Discussion section.

We thank the reviewer for this comment. Regarding why ACE2-Ig-105/106 has significantly lower yield than that of ACE2-Ig-95, we think the following two factors might contribute to this. First, ACE2-Ig-95 is a homodimer protein. In contrast, ACE2-Ig-105/106 is a tetramer protein, which is a homodimer of two heterodimers (heavy chain + light chain). This could result in higher chance of failure to produce fully assembled ACE2-Ig-105/106 tetramers. Second, the more complicated ACE2-Ig-105/106's tetramer configuration might cause more protein-folding stress in the cells (Hetz *et al. Nat Rev Mol Cell Biol* 2020) and thus lower the yield of the protein. This is also consistent with the observation that the ACE2-Ig-105/106 stable cell pool has a lower growth rate than the ACE2-Ig-95 stable cell pool (Figure 3A). We have now added these discussions to the Discussion section (lines 424-429, Marked Up version).

2) In BALB/c mice, ACE2-Ig-95 has a longer half-life compared to ACE2-Ig-105/106 (43.0 {plus minus} 1.8 vs 20.4 {plus minus} 0.3 h) (lines 225-226; Figure 3C-3F); however, similar efficacy in reducing pulmonary viral load can be observed, especially via the *i.n.* route (Figure 3H).

We thank the reviewer for pointing out this. Because ACE2-Ig-105/106 has ~2-fold shorter *in vivo* half-life (Figure 3C-F) but >5-fold higher *in vitro* neutralization potency against WHU01 than ACE2-Ig-95 (Figure 1B), the result that these two proteins have similar *in vivo* efficacy (Figure 3H) is actually consistent with the half-life and neutralization potency data. On the other hand, it's worth noting that ACE2-Ig-105/106 was found significantly more effective when administered via the *i.n.* route (Figure 3H). Our hypothesis for this observation is that the large sized ACE2-Ig-105/106 protein (~430 kDa) has poor bioavailability and distribution to the lungs and administration via the *i.n.* route bypasses the bioavailability issue of the protein. For the above

reasons, i.n. administration was then opted for in the following *in vivo* efficacy studies, and again similar *in vivo* efficacies were observed for the two proteins (Figure 4 and 5).

3) Compared to the positive control (Etesevimab), ACE2-Ig-95 and ACE2-Ig-105/106 exhibit stronger neutralization activities (Figure 1 and Figure 2); however, the animals treated with the positive control (Etesevimab) seem to display better therapeutic effects than those treated with ACE2-Ig-95 or ACE2-Ig-105/106 at the same dosage (25 mg/kg) (Figure 4 and Figure 5). Additionally, among the treatment dosages, the 10 mg/kg dose of ACE2-Ig-95 or ACE2-Ig-105/106 provides more effective protection against SARS-CoV-2 infection, including survival rate, pulmonary viral load, and histopathological lesions, et al. (Figure 4 and Figure 5). How can this observation be explained?

We thank the reviewer for these insightful comments.

Regarding the efficacy of Etesevimab vs ACE2-Ig-95 and ACE2-Ig-105/106, our *in vitro* and *in vivo* data are highly consistent. Figures 1, 2, and S3 show *in vitro* neutralization data for diverse SARS-CoV-2 variants while Figures 4 and 5 show *in vivo* protection data only for the SARS-CoV-2 Hong Kong isolate (Hong Kong/VM20001061/2020). Although ACE2-Ig-95 and ACE2-Ig-105/106 have significantly better average IC₅₀ than Etesevimab does against diverse SARS-CoV-2 variants (Figure 2), they are similarly effective as Etesevimab in neutralizing an early isolate “D614G” (Figure S3A). In addition, to further strengthen the manuscript and further confirm the neutralization breadth of our ACE2-Ig proteins against diverse SARS-CoV-2 variants, especially against the extremely immune-evasive BQ and XBB subvariants, we have now performed additional *in vitro* neutralization assays using multiple authentic SARS-CoV-2 variants and included the data in the revised manuscript as a new figure (Figure 6). Figure 6A shows that Etesevimab is actually more effective than both ACE2-Ig-95 and ACE2-Ig-105/106 against the early isolate WHU01. The SARS-CoV-2 Hong Kong isolate (Hong Kong/VM20001061/2020; GISAID: EPI_ISL_412028) used in the mouse experiments (Figure 4 and 5) was collected in the very beginning of the pandemic (collection date: 2020-01-22) thus is expected to behave similar/identical to the WHU01 isolate. Therefore, the *in vitro* data (Figure 6A and S3A) are actually highly consistent with the *in vivo* data (Figure 4 and 5). However, our *in vitro* data in Figure 2 and 6 do demonstrate that, in contrast to monoclonal antibodies, ACE2-Ig is very potent, exceptionally broad, and difficult to escape, thus is a good drug candidate against diverse ACE2-utilizing coronaviruses.

Then as the reviewer pointed out, ACE2-Ig-95 or ACE2-Ig-105/106 at 10 mg/kg dose indeed provides more effective protection than the same protein at 25 mg/kg in terms of histopathological lesions, clinical score, and survival rate (Figures 4 and 5). This might be associated with ACE2's peptidase activity. Because ACE2 is an endogenous protein with broad endogenous substrate specificity and directly participate in the renin angiotensin system, higher dose of peptidase-active ACE2-Ig proteins might be associated with some side effects that compromise the protein's antiviral therapeutic effect. More thorough toxicology studies are necessary in the future and would give us more confirmative answer to this. Nevertheless, we now have added this point to the Discussion section (lines 396-399, Marked Up version). We believe that the updated discussion on ACE2's peptidase activity in the Discussion section (lines 395-403, Marked Up version) is more thorough and significantly improved over the original version.

4) It would be beneficial for the authors to review the manuscript for consistency in grammar and word usage. For example, a uniform description for the unit of protein concentration should be used throughout the manuscript, either "gram/liter" (line 50) or "g/L" (lines 213, 395, 499).

We thank the reviewer for this comment and have done so accordingly for the manuscript.

Responses to Reviewer #2 (reviewer comments in black; author responses in blue)

Reviewer #2 (Comments for the Author):

This study reports the development of two human ACE2 decoy proteins ACE2-Ig-95 and ACE2-Ig-105/106 and demonstrated these two proteins have a potent in vitro neutralization activity against tested SARS-CoV-2 variants. In lethal human ACE2 transgenic mice, the proteins could significantly decrease the viral load in lung tissue and alleviate the severity and mortality. In comparison with previously reported ACE2-Ig proteins with residue mutations, the study showed that the ACE2 mutations may lose the neutralization potency against the SARS-CoV-2. Furthermore, the study demonstrated the high production capacity in the laboratory. The study was well designed and the data is solid, which provide alternative route for SARS-CoV-2 treatment.

We appreciate the reviewer's careful evaluation and positive comments on our study.

Minor comments:

1. In addition to the SARS-CoV-2 variants used in in vitro study, authors please indicate clearly which strains and how many dose used in different animal challenge studies.

We thank the reviewer for this comment. We have done so accordingly, and the virus strain and dose information now can be found on lines 594-597 and lines 608-609 of the main text (Marked Up version).

2. Authors should discuss why the 10 mg/kg is more potent (Figure 4)?

We thank the reviewer for these insightful comments. ACE2-Ig-95 or ACE2-Ig-105/106 at 10 mg/kg dose indeed provides more effective protection than the same protein at 25 mg/kg in terms of histopathological lesions, clinical score, and survival rate (Figures 4 and 5). This might be associated with ACE2's peptidase activity. Because ACE2 is an endogenous protein with broad endogenous substrate specificity and directly participate in the renin angiotensin system, higher dose of peptidase-active ACE2-Ig proteins might be associated with some side effects that compromise the protein's antiviral therapeutic effect. More thorough toxicology studies are necessary in the future and would give us more confirmative answer to this. Nevertheless, we now have added this point to the Discussion section (lines 396-399, Marked Up version). We believe that the updated discussion on ACE2's peptidase activity in the Discussion section (lines 395-403, Marked Up version) is more thorough and significantly improved over the original version.

June 6, 2023

Dr. Guocai Zhong
University of Massachusetts Chan Medical School
Biochemistry and Molecular Biotechnology
364 Plantation Street LRB-815
Worcester, MA 01605

Re: Spectrum01100-23R1 (Broadly effective ACE2 decoy proteins protect mice from lethal SARS-CoV-2 infection)

Dear Dr. Guocai Zhong:

Your manuscript has been accepted, and I am forwarding it to the ASM Journals Department for publication. You will be notified when your proofs are ready to be viewed.

Sincerely,

Tao Deng
Editor, Microbiology Spectrum
